# A Compact Mu-Near-Zero Metamaterial Integrated Wideband High-Gain MIMO Antenna for 5G New Radio Applications

**DOI:** 10.3390/ma16041751

**Published:** 2023-02-20

**Authors:** Md. Mhedi Hasan, Mohammad Tariqul Islam, Sharul Kamal Abdul Rahim, Touhidul Alam, Hatem Rmili, Ahmed Alzamil, Md. Shabiul Islam, Mohamed S. Soliman

**Affiliations:** 1Department of Electrical, Electronic and Systems Engineering, Faculty of Engineering and Built Environment, Universiti Kebangsaan Malaysia (UKM), Bangi 43600, Selangor, Malaysia; 2Department of Information and Communication Technology (ICT), Faculty of Engineering, Comilla University, Cumilla 3506, Bangladesh; 3Wireless Communication Centre, Universiti Teknologi Malaysia, Skudai 81310, Johor, Malaysia; 4Pusat Sains Ankasa (ANGKASA), Institut Perubahan Iklim, Universiti Kebangsaan Malaysia (UKM), Bangi 43600, Selangor, Malaysia; 5Electrical and Computer Engineering Department, Faculty of Engineering, King Abdulaziz University, Jeddah 21589, Saudi Arabia; 6K. A. CARE Energy Research and Innovation Center, King Abdulaziz University, Jeddah 21589, Saudi Arabia; 7Electrical Engineering Department, College of Engineering, University of Hail, Hail 81481, Saudi Arabia; 8Faculty of Engineering, Multimedia University (MMU), Cyberjaya 63100, Selangor, Malaysia; 9Department of Electrical Engineering, College of Engineering, Taif University, P.O. Box 11099, Taif 21944, Saudi Arabia; 10Department of Electrical Engineering, Faculty of Energy Engineering, Aswan University, Aswan 81528, Egypt

**Keywords:** metamaterial MIMO antenna, Mu-near-zero metamaterial, wideband, high gain, 5G NR bands, gain improvement, compact size

## Abstract

This article demonstrates a compact wideband four-port multiple-input-multiple-output (MIMO) antenna system integrated with a wideband metamaterial (MM) to reach high gain for sub-6 GHz new radio (NR) 5G communication. The four antennas of the proposed MIMO system are orthogonally positioned to the adjacent antennas with a short interelement edge-to-edge distance (0.19λ_min_ at 3.25 GHz), confirming compact size and wideband characteristics 55.2% (3.25–5.6 GHz). Each MIMO system component consists of a fractal slotted unique patch with a transmission feed line and a metal post-encased defected ground structure (DGS). The designed MIMO system is realized on a low-cost FR-4 printed material with a miniature size of 0.65λ_min_ × 0.65λ_min_ × 0.02λ_min_. A 6 × 6 array of double U-shaped resonator-based unique mu-near-zero (MNZ) wideband metamaterial reflector (MMR) is employed below the MIMO antenna with a 0.14λ_min_ air gap, improving the gain by 2.8 dBi and manipulating the MIMO beam direction by 60°. The designed petite MIMO system with a MM reflector proposes a high peak gain of 7.1 dBi in comparison to recent relevant antennas with high isolation of 35 dB in the n77/n78/n79 bands. In addition, the proposed wideband MMR improves the MIMO diversity and radiation characteristics with an average total efficiency of 68% over the desired bands. The stated MIMO antenna system has an outstanding envelope correlation coefficient (ECC) of <0.045, a greater diversity gain (DG) of near 10 dB (>9.96 dB), a low channel capacity loss (CCL) of <0.35 b/s/Hz and excellent multiplexing efficiency (ME) of higher than −1.4 dB. The proposed MIMO concept is confirmed by fabricating and testing the developed MIMO structure. In contrast to the recent relevant works, the proposed antenna is compact in size, while maintaining high gain and wideband characteristics, with strong MIMO performance. Thus, the proposed concept could be a potential approach to the 5G MIMO antenna system.

## 1. Introduction

The growing advancement of high-speed modern communication systems, such as fifth generation (5G) communications, has triggered a new area of research for enhancing the system capacity and achieving multipath communications [1,2,3]. The multiantenna system, called multiple-input-multiple-output (MIMO) technology, is the core technology of the 5G networks because of its potential applications in industry 4.0 and beyond, including smart health, smart city, smart transportation, smart home, wearable devices, and more [4,5]. The MIMO system can effectively improve the data rate, signal quality, and channel capacity with limited bandwidth and power requirements compared to the single antenna system [6,7]. Additionally, it can also mitigate the fading effects and achieve diversity characteristics [2,7]. 

At present, the most common 5G sub-6 GHz new radio (NR) spectrums are being used worldwide owing to their widespread adoption and seamless coverage [1,8]. Various wireless terminal and access point applications require compact wideband and high gain antennas to enhance the system performance and coverage area while reducing the antenna deployment cost. Thus, researchers worldwide are concentrating more on developing compact wideband MIMO antenna technology with high isolation and high gain [9,10,11,12,13]. The microstrip patch antenna is a potential candidate in MIMO systems due to its compactness and low cost, although it has limited gain and bandwidth [14,15,16]. However, the mutual coupling between MIMO components is a major concern as the antenna radiation and diversity performance are deteriorated by the harmful mutual coupling effects. Therefore, research on improving the bandwidth, isolation, and gain of MIMO antennas is a major challenge in the research community [5,9,17,18]. To mitigate the MIMO antenna’s shortcomings, particularly its low gain, narrow bandwidth, and mutual coupling between MIMO components, various techniques have been introduced, such as parasitic elements [9,19], electromagnetic bandgap (EBG) [5,20,21], defected ground structure (DGS) [22], metamaterial (MM) [12,17,23,24,25] and metamaterial absorber (MA) [11]. The EBG structure has been positioned between the MIMO components along with DGS [5,26], reducing surface wave propagation. Hence, the isolation between MIMO antennas is significantly improved; however, these designs demonstrated limited MIMO elements, operating bandwidth (BW), and gain enhancement. 

The metamaterial is a promising technique to improve the MIMO antenna’s performance, particularly in terms of its gain and isolation between the MIMO elements [12,17,23,24]. It is an artificially engineered, unique material that can modulate electromagnetic (EM) waves, enhancing the antenna performance [27,28]. In [9,23], the authors presented a low-profile two-element MIMO antenna incorporating the metasurface (MS) or parasitic components around the antenna radiators on a single substrate. These antennas are low profile and highly isolated; however, they have low gain, limited bandwidth, and MIMO elements. Similarly, a square parasitic plate-based 2-elements wideband MIMO antenna with a maximum gain of 6.45 dB is developed in [29]. In addition, in [12], the MS layer has been placed directly on the antenna patch without any air gap in order to confirm the low profile with high gain and isolation. However, the system is large in size and has a poor operational bandwidth, limiting its wide applicability in 5G NR sub-6 GHz applications. On the other hand, the array of metamaterial (MM) unit cells has been introduced between the two MIMO components to attain high isolation between two antenna radiators [11,30,31] with miniaturized dimensions. Nonetheless, the number of MIMO elements and gain improvement are the major limitations of these reported MIMO antennas. Moreover, the antennas in [11,30], operated in very narrow bandwidths with resonance frequencies of 5.5 and 5.8 GHz, respectively, which do not cover the 5G NR spectrums (n77/n78/n79). In addition, an E-shaped metamaterial-inspired 2-elements MIMO antenna with enhanced isolation and bandwidth is reported in [32]. However, it lacks MIMO diversity analysis and has a high inter-element edge-to-edge gap.

In various MIMO antenna designs, the metamaterial is laid above [17,33,34,35,36] or beneath [13,37] the antenna radiator with an air gap. In these instances, the MM performs as a superstrate or reflector that manipulates the gain, isolation, and bandwidth of the antenna. A closely loaded 2-elements dual-band MIMO antenna is developed in [17], where a double-layered MS superstrate is proposed with a height of 11 mm to improve the isolation between two MIMO antennas. However, this communication highlighted the limited frequency range of its operation at the dual resonances, with only two MIMO elements, as well as a limited gain enhancement at the lower band (0.9 dBi with MS). In [33], a multi-layered MS superstrate with epsilon negative characteristics is presented to enhance the isolation between the antenna elements. Although high isolation is achieved by this multilayer superstrate technique, the antenna has a limited bandwidth and a complex design because of its multilayer structure. The bandwidth and gain enhancement of a two-port MIMO array antenna using MM superstrate is proposed in [34]. However, this antenna operates at 5.65–6.4 GHz with a large dimension, which limits its applicability in 5G bands. In addition, another MM superstrate MIMO antenna with excellent gain and isolation is proposed in [35] for WLAN applications, although the maximum gain improvement is only 1.47 dBi, with a narrow bandwidth and larger dimension in respect to the working spectrum and MIMO elements. In [13], the MS reflector is positioned at the bottom of the two-element MIMO antenna with DGS for enhancing the antenna gain. However, it has only two antenna elements and a large air gap (15 mm) between the antenna and the MS reflector. Moreover, the proposed MS reflector size is greater than the designed MIMO antenna, limiting its compactness. Similarly, the authors in [37] proposed another MS reflector-based multiband MIMO antenna for enhancing gain and isolation, where they proposed only two element MIMO antenna, incorporating a larger MS reflector than the antenna size, which hinders the compactness of the antenna. The above research reveals that most contemporary MIMO antennas are designed to enhance isolation, but the gain enhancement and isolation improvement of the miniaturized MIMO antenna using MM for 5G NR spectrums is less prevalent. Thus, designing a MM-based high gain four-port MIMO patch antenna with miniature dimensions, wide coverage (n77/n78/n79 band), high isolation, and excellent diversity properties is a huge challenge for 5G communication in the sub-6 GHz NR bands. 

Accordingly, this work presents a compact, wideband and high gain 4-port MIMO (4 × 4 MIMO) antenna incorporated with a wideband MM reflector for sub-6 GHz 5G applications. The designed MIMO system has perceptible characteristics of high gain while maintaining wideband coverage, and miniaturized dimensions with a short edge-to-edge gap of 0.19λ_min_ between the adjacent antenna radiators. Moreover, high isolation between the MIMO components is achieved using a wideband MM with an outstanding MIMO performance, confirming the feasibility of the developed antenna in 5G NR MIMO communication systems. A metal post-enclosed DGS with a microstrip feed line is devised in the initial antenna to alleviate the limited covering frequency. The four identical single antennas are orthogonally arranged to the adjacent antennas in the proposed MIMO system. A wideband Mu-near-Zero (MNZ) MM is developed in the MIMO system to improve the system performance, most notably, the gain and isolation. Moreover, the proposed metamaterial reflector (MMR) enhances the directivity of the MIMO beam pattern by shifting the main beam direction to the top of the radiators. The designed MIMO configuration is further investigated in terms of the envelope correlation coefficient (ECC), diversity gain (DG), channel capacity loss (CCL), and multiplexing efficiency (ME) to assess its applicability in the 5G n77/n78/n79 bands’ MIMO applications. These investigations are necessary to perform diversity analysis, which confirms the MIMO antenna’s quality and its practical applicability in 5G networks. Finally, the proposed MIMO structure is fabricated and examined. The electromagnetic analysis simulator platform CST studio suite 2019 has been utilized to optimize and simulate the proposed MIMO system.

## 2. MM-Based Reflector

Metamaterials have potential applications in 5G communication systems for enhancing antenna performance due to their engineerable characteristics, including permittivity, permeability, and refractive index [24,38,39,40,41]. In this study, a unique wideband metamaterial reflector with MNZ characteristics is designed to improve the antenna performance, covering the 5G NR n77/n78/n79 frequency band. The proposed unit cell structure is developed utilizing a coupled ring encased double U-shaped split ring resonator which complies with the MM sub-wavelength standard to achieve an effective performance [42]. The purpose of the proposed design is to cover the 5G NR n77/n78/n79 bands with a compact size including Mu-near-Zero (MNZ) and epsilon negative (ENG) characteristics. Initially, the MM design target is to reach the 5G NR spectrums with wideband properties; hence, the estimated initial dimensions of the proposed structure are opted at the frequency of 3.5 GHz. The chosen compact unit cell size (L) of 10 mm implies that L = λ_L_/8.57 (where the wavelength λ_L_ = 85.7 mm at the targeted frequency of 3.5 GHz), which is compact enough to satisfy the metamaterial sub-wavelength requirement and, therefore, the MM effective response can be achieved. A couple ring enclosed double U shape (CREDUS) unique resonator is developed on 1.6 mm height low-cost FR-4 printed material (
${𝜀}_{r}=4.3$ and tan𝛿=0.025 with a compact size of 0.11λ_min_ × 0.11λ_min_ × 0.02λ_min_ at 3.4 GHz. The optimized unit cell geometry is comprised of double U shape rings that are enclosed to the two coupled square complementary split rings, as revealed in Figure 1a. Figure 1a displays the detailed MM reflector design parameters, including a unit cell magnified image and a snapshot of the developed prototype. The 6 × 6 arrays of unit cells have been utilized to develop an MM reflector with a dimension of 60 × 60 mm^2^ (M_W_ × M_L_), as indicated in Figure 1a. The proposed CREDUS resonator is designed and numerically analyzed utilizing an adaptive tetrahedral mesh-based frequency domain solver on the EM simulator platform, CST studio suite 2019. In the simulation process, the electric field and magnetic field are parallel to the MM structure, whereas the EM waves are directed on top of the MM. Figure 1b shows the simulated scattering parameters (S-Parameters) of the reported MMR. It can be seen that the designed MM has a wide −10 dB bandwidth (S_21_) of 3.4 to 5.2 GHz and resonates at 4.61 GHz, showing excellent reflection qualities in the spectrum of 3.4–5.2 GHz [13,37]. Notably, the
$\left|S_{21}\right|$ and $\left|S_{11}\right|$ parameters (Figure 1b) show wide band-stop properties between 3 GHz and 5.7 GHz, which cover the 5G n77/n78/n79 frequency spectrum. Moreover, the wide linear high
$\left|S_{11}\right|$ parameters of the designed MMR help in improving the antenna gain when employing at a proper gap owing to the cavity effect [24,35,43].

The effective parameters, such as the permittivity and permeability of the reported MM, have been investigated to further analyze the MM behavior in depth. An improved retrieval method can be utilized to retrieve the MM’s effective parameters using a scattering matrix [44]. The following S-parameters, Equations (1) and (2), are used to determine the effective impedance, *z*, and refractive index, *n* [27].

Reflection coefficient:(1)$$S_{11}=\frac{R_{01}\left(1-e^{i2nk_{0}d}\right)}{1-R^{2}{}_{01}e^{i2nk_{0}d}}$$

Transmission coefficient:(2)$$S_{21}=\frac{(1-R^{2}{}_{01})\;e^{ink_{0}d}}{1-R^{2}{}_{01}e^{i2nk_{0}d}}$$
where $R_{01}=\frac{z-1}{z+1}$, d is the substrate thickness and *k_0_* indicates the free space wave number.

From Equations (1) and (2), it can be obtained:

Impedance: (3)$$z=\pm \sqrt{\frac{{\left(1+S_{11}\right)}^{2}-S^{2}{}_{21}}{{\left(1-S_{11}\right)}^{2}-S^{2}{}_{21}}}$$

Refractive index:(4)$$n=\frac{1}{K_{0}d}\;\{\left[{\left[\ln\left(e^{ink_{0}d}\right)\right]}^{\prime \prime }+2m\pi \right]-i\;{\left[\ln\left(e^{ink_{0}d}\right)\right]}^{\prime }\}$$
where (.)″ and (.)′ represent the imaginary and real parts of the operator, respectively, and m is the integer denoting the branch index for the real part of n. The following Equations (5) and (6), can be used to calculate the MM’s effective permittivity (*ε*) and permeability (*µ*), respectively:

Relative permittivity:(5)$$\varepsilon =\frac{n}{Z}$$

Relative Permeability:(6)μ=nz

The built-in post-processing method of the EM simulator (CST) is used in this work to retrieve the designed MM’s effective parameters. Figure 2a–c reveals the proposed wideband MMR’s effective parameters. From the permittivity (*ε*) plot, portrayed in Figure 2a, a negative value is noted in the frequency range of 4.6 to 5.35 GHz. Interestingly, as depicted in Figure 2b, the developed MMR exhibits a near-zero positive permeability value (real) spanning the proposed antenna working spectrum, confirming Mu-near-Zero (MNZ) and epsilon negative (ENG) characteristics. From the following Maxwell Equations (7) and (8), it is obvious that the material with near-zero characteristics reduces the near-field coupling between the magnetic and electric fields [35,45].
(7)∇×E=iωμH
(8)∇×H=−iωεE
where H and E denote the magnetic and electric fields, respectively, at frequency ω. In addition, the developed MMR exhibits a negative index property, as shown in Figure 2c, implying this could play a vital role in improving the antenna’s performance [46]. Thus, the proposed wideband MMR with MNZ and a negative index property can be used to improve the gain and isolation in the wideband MIMO antenna technology [18,24,35,47,48].

## 3. MIMO Antenna Design and Analysis with MM

Initially, a fractal slotted patch and metal post-enclosed defected ground structure (DGS) microstrip antenna is designed, covering the 5G NR frequency band. Then, a 4 × 4 MIMO antenna is designed using four identical single antennas, which are arranged orthogonally to the neighboring antennas. Finally, the 6 × 6 array of the developed MM is positioned below the designed MIMO antenna for enhancing the MIMO system performance. The proposed antenna is developed on a low-cost 1.6 mm height FR-4 printed substrate $(\varepsilon_{r}=4.3$). The proposed antenna structure is designed and optimized using the EM simulator CST studio suite 2019.

### 3.1. Single Antenna Design

Figure 3a demonstrates the proposed single antenna design process with a compact size of 30 × 30 mm^2^. The initial antenna is comprised of a rectangular radiator, which is excited by a 50 Ω transmission line, as indicated in Figure 3a (first step). The initial radiating patch and transmission line dimensions are determined using standard mathematical equations [49]. In the initial phase, the developed antenna patch resonates at 7 GHz with a low impedance bandwidth. The antenna back copper and feeding position is optimized in the second stage, covering the n79 band with increasing impedance matching [50]. In this phase, the resonant frequency shifted significantly to the lower range. The third and fourth stage incorporates plus-shape fractal etching on the radiating patch of the antenna to shift the resonant frequency and achieve miniaturization and good impedance matching. The plus shape slot shifted the antenna operating frequency to a lower range by generating an extra capacitance effect. In the first fractal iteration (step-3 in Figure 3a), the antenna resonates at 5.1 GHz, covering 4.6–5.6 GHz, as demonstrated in Figure 3b. The first plus-shape etching on the patch shifted the resonance frequency to the lower range with the increasing impedance matching. In step four (second iteration), the antenna offers good impedance matching and resonated at 4.8 GHz, as exhibited in Figure 3b. In the final stage, a metal stub is incorporated into the antenna backplane and the feeding position is shifted to optimize the antenna’s operating frequency and impedance matching. The metal stub on the DGS and optimized feeding position yields a wide impedance BW of 2.43 GHz (3.37 to 5.8 GHz) with excellent impedance matching, covering Sub-6 GHz NR n77/n78/n79 frequency spectrums, as demonstrated in Figure 3a,b. The single antenna detailed design specifications are: *A*_L_ = 16, *A*_W_ = 30, *P*_w_ = 15, *P*_L_ = 10, *g*_L_ = 6.3, *f*_L_ = 11, *f*_W_ = 1.5, *s*_1_ = 4.5, *s*_2_ = 6, *s*_3_ = 3.25, and *c* = 12.75 (unit: mm).

The final optimized antenna is fabricated to assess the real-world performance of the developed prototype. Low-cost FR-4 material was used in this proposed prototype to reduce the fabrication cost in the industry. Figure 4a illustrates a snapshot of the developed single antenna prototype and Figure 4b reveals the simulated and measured reflection coefficient (S_11_) curves. The simulated $\left|S_{11}\right|$ indicates a wide bandwidth from 3.37 to 5.8 GHz, covering the NR n77/n78/n79 5G frequency spectrums. The simulation findings are supported by the measured results, as indicated in Figure 4b, where the observed $\left|S_{11}\right|$ shows a value of −10 dB over the target band of 3.35–5.8 GHz. However, the numerical and experimental results demonstrate slight discrepancies because of manufacturing defects, measurement tolerance, soldering effect, and wire and SMA connection loss.

### 3.2. MIMO Antenna Design with MM

The proposed compact wideband four-port MIMO antenna schematic architecture is displayed in Figure 5a, where four single antennas are positioned orthogonally to one another. The four excitation ports of the proposed antenna are positioned at the four edges, with a 50 Ω microstrip feedline. The edge-to-edge distance between the MIMO antenna elements is 0.19λ_min_, which is more compact than the recently developed antennas. Furthermore, the DGS with a metal post of the MIMO components is arranged in the same manner as the unit antenna. The proposed model optimization and numerical investigation were conducted utilizing the EM-based simulator, CST studio suite 2019. The designed MIMO antenna is realized on a low-cost FR-4 printed material with a compact dimension of 60 × 60 × 1.6 mm^3^. Figure 5b depicts a fabricated MIMO prototype with a SMA connector. Figure 6 reveals the simulated scattering parameter curves of the developed MIMO antenna, which resonates at 3.55, 4.5, and 5.2 GHz with a wide −10 dB impedance bandwidth of 3.3 GHz to 5.7 GHz, covering the 5G (n77/n78/n79) frequency spectrum. Although the working frequency is somewhat shifted to the lower range, owing to the MIMO near-field coupling effects, the impedance bandwidth shows similar results. The 5G sub-6 GHz NR bands are nicely covered by this adjustment. Figure 6 also shows the developed MIMO antenna’s isolation curves (transmission coefficients) without MM. Mutual coupling is induced in the MIMO systems by surface wave and space wave coupling between the antenna components. The minimum isolation is noted at approximately 9.1 dB between MIMO elements 1 and 3. This low isolation is realized due to strong space wave coupling and surface wave coupling between antenna 1 and 3. The maximum 4.3 dBi gain is achieved by the designed MIMO system due to the near-field mutual coupling effect between the MIMO antennas. Therefore, an effective isolation and gain enhancement approach is required to improve the MIMO antenna isolation and gain.

In this work, a unique MNZ wideband metamaterial reflector, with a dimension of 60 × 60 × 1.6 mm^3^ (6 × 6 array of designed unit cells, similar size to MIMO antenna), is proposed to enhance the isolation and gain of the developed MIMO antenna. Figure 7a shows the developed MIMO antenna with an MMR (side view) and its simulation approach. The developed MMR is positioned behind the design 4-elements MIMO antenna using two 12.5 mm height nylon spacers, as presented in Figure 7a. The distance between the MM reflector and the antenna is a key issue for reaching the maximum broadside performance while generating additional resonances by interfering favorably with the MM reflected radiation [17,24,35,51]. Moreover, the MM inclusion in the antenna systems manipulates the antenna surface wave to enhance the antenna performance [17,38]. Thus, the proposed MMR improves the antenna beam directivity because of the cavity effect and reduces the mutual coupling due to the surface wave suppression. Various simulations have been conducted to ascertain the optimal distance of the MM from the antenna for reaching high isolation and high gain. As the MM reflector is very near to the MIMO antenna (h = 12.5 mm, which is 0.14λ_min_ at 3.25 GHz), it has a slight influence on the antenna matching characteristics. Thus, the only antenna ground height and feeding point are slightly optimized to confirm the matching characteristics, while the other parameters remain unchanged.

## 4. Results and Discussion

Figure 7b exhibits the images of the fabricated optimized MIMO antenna with a MM reflector and the near field (SATIMO) measurement setup to verify the numerical findings. The fabricated MIMO prototype is positioned on top of the MM reflector using two nylon spacers, at the two opposing vertical positions of the prototype, for experimental analysis, as presented in Figure 7b. In the reflection coefficient measurement process, only one antenna is energized, while the remaining three antennas are terminated using a 50 Ω terminator (Figure 7b). Conversely, the two antennas of the MIMO system are excited simultaneously for the isolation measurement, while the remaining two ports are terminated using a 50 Ω terminator.

### 4.1. Reflection Coefficients and Isolation

Figure 8a,b presents the simulated and observed reflection coefficients (S_11_ and S_22_) of the MIMO antenna with and without MM. As indicated in Figure 8a,b, the developed MIMO systems, both with and without MM, have similar impedance bandwidths of −10 dB. The developed MM-based MIMO antenna offers a broad impedance band of 3.25–5.6 GHz (55.2% fractional bandwidth at 4.26 GHz). The MM effects have caused a little shift in the operational frequency; however, the 5G sub-6 GHz NR bands are well covered. The measured impedance, BW 2.54 GHz (3.26–5.8 GHz), is noted for the initially developed MIMO antenna and 2.51 GHz (3.2–5.71 GHz) after employing the metamaterial. Figure 8c,d shows the simulated and measured isolation curves of the developed MIMO antenna (without and with MM), demonstrating excellent correlation with the numerical data. From the isolation plots in Figure 8c,d, it is evident that when the MM is incorporated into a MIMO system, the isolation between the MIMO antennas is noticeably enhanced. When the proposed MMR is positioned in the vicinity of the MIMO system, the coupling current is loaded in the MM, which reduces the near-field coupling effects between the antennas. The least isolation between the adjacent (S_12_/S_14_) and diagonal (S_13_/S_24_) antennas in the 5G NR bands is >12 dB, indicating a minimum 3 dB improvement in the isolation; the highest isolation is 35 dB. The experimental S-parameters are almost identical to the simulated results, confirming the possible applications in the 5G NR MIMO antenna system. The measured curves demonstrated in Figure 8 are slightly shifted from the simulated one due to the measurement tolerance, manufacturing, and assembly tolerance.

To clarify the isolation mechanism of the developed MM reflector, the surface current distribution and electric field distribution of the designed MIMO and the MIMO with the MM antenna are investigated at 4.5 GHz, as depicted in Figure 9 and Figure 10. A significant coupling current is transferred to the nearest antennas, increasing the near-field mutual coupling effect between MIMO antennas (Figure 9a). After using the MM reflector, the near-field coupling current is concentrated in the MM unit cells, where the current traverses in an antiparallel fashion in one half of the ring and the following rings, as shown in Figure 9 (b). In addition, the residual edge currents of the adjacent MIMO antennas are in an antiparallel cancelation fashion. Therefore, the near-field coupling current between the adjacent antennas is suppressed, resulting in high isolation. Figure 10 shows the electric field (E-field) distributions for the MIMO and the MIMO with the MM antenna, where antenna 1 is excited. A strong coupled electric field is found in antennas 2, 3, and 4 of the MIMO system when the MM is absent (Figure 10a). After employing the MM reflector, the coupled E-field distribution on the MIMO antennas (antenna 2, 3, and 4) is minimized significantly, as shown in Figure 10b.

### 4.2. Gain and Efficiency

The radiation characteristics of the proposed MIMO antenna prototypes (with and without a MM reflector) are measured with the SATIMO near-field laboratory set-up. Figure 11a depicts the measured gain for the proposed MIMO prototype in comparison to the results of the simulations, demonstrating good accordance. It can be found that the simulated maximum gain of the designed 4-elements MIMO antenna (without MM) is 4.3 dBi, which is lower than the relevant reported antennas. Thus, the proposed wideband MMR is equipped behind the MIMO system, enhancing the antenna radiation characteristics and yielding a high gain of 7.1 dBi [37,51]. The MNZ- MMR manipulates the antenna surface waves by experiencing favorable interference, resulting in supplementary resonances and enhancing the antenna radiation qualities [20,38]. The inclusion of the MM in the developed MIMO system improves the peak realized gain, from 4.3 dBi to 7.1 dBi with a gain enhancement of 2.8 dBi, which is also validated experimentally, in Figure 11a. The simulated and measured gains (realized) of the developed MIMO prototype (with and without MM) are almost consistent. Although there is a little disparity between the experimental and simulated results, this is mostly due to manufacturing flaws, antenna and MS misalignment, terminated port reflection, and measurement tolerance. The developed MIMO system has a maximum observed gain of 5.1 dBi (without MM) and 7.75 dBi in the presence of a MM. It is important to note that the proposed MM-based MIMO antenna has a high gain for both the modelling and experimental environments. Figure 11b reveals the simulated and observed total efficiency of the developed MM-based MIMO antenna. As shown in Figure 11b, the reported MIMO antenna provides excellent total efficiency, of up to 74% in the operating frequency. The measured maximum total efficiency is 71.5%, which is very near to the simulated efficiency. The antenna efficiency is influenced by mutual coupling and material losses. The near-field coupling effect is enhanced by the surface wave and space wave coupling among the MIMO elements, which modifies the input impedance and radiation characteristics of the MIMO antennas. Moreover, the lossy FR-4 materials and copper loss contribute to the dissipation of the input power in the proposed antenna systems. These factors may have significant influences on degrading the antenna efficiency [52,53]. Whilst the mutual coupling between MIMO components affects the antenna efficiency, the antenna efficiency might be improved by enhancing the isolation among the MIMO components. Moreover, minimizing material losses and impedance mismatches may enhance antenna efficiency. Furthermore, antenna radiator engineering might be another possible strategy to improve the antenna efficiency [54].

### 4.3. Radiation Patterns

Figure 12a,b shows the proposed MIMO antenna radiation patterns at 4.5 GHz. The MM reflector in the MIMO system considerably enhanced the antenna beam directivity in both planes (E- plane and H-plane). Moreover, the main beam direction is shifted to the top of the patch by 60° and 6° in the x-z and y-z planes, respectively, while utilizing a MM reflector. Notably, the designed MM-based MIMO antenna achieves a unidirectional beam pattern with low back- and side-lobe values. The fabricated MIMO prototype is tested in the SATIMO near-field system laboratory under both MM and MM-free conditions to verify the simulated radiation pattern. The experimental radiation properties have been extracted from port 1, while the remaining ports are terminated by the 50 Ω terminators. Figure 12a,b plots the observed E-plane and H-plane radiation characteristics. A close match has been found between the observed and simulated radiation patterns. The experimental results slightly fluctuate from the simulated plots due to manufacturing defects, the misalignment of components, the measurement tolerance, the terminated port reflection, the proximity to various electronic devices, commercial grade spacer, cable loss, and various connection-related loss. Nevertheless, despite these constraints, the proposed MIMO prototype works well, with excellent agreement between the measurement and simulation, considering it suitable for 5G NR applications.

### 4.4. Envelope Correlation Coefficient (ECC)

The envelope correlation coefficient (ECC) among the MIMO elements is a crucial indicator for deciding the applicability of the MIMO antenna in the 5G communication systems. The ECC between the MIMO antennas determines the correlation of the individual elements in their independent performance. The ideal ECC value of the MIMO antenna is zero, but <0.5 is acceptable [10,11]. The ECC of the proposed MIMO antenna can be determined from the S-parameters or the antenna radiation patterns, utilizing Equations (9) and (10), respectively [55].
(9)$$ECC=\frac{{\left|S_{ii}*S_{ij}+S_{ji}*S_{jj}\right|}^{2}}{\left(1-{\left|S_{ii}\right|}^{2}-{\left|S_{ij}\right|}^{2}\right)\left(1-{\left|S_{ji}\right|}^{2}-{\left|S_{jj}\right|}^{2}\right)}$$
(10)$$ECC=\frac{{\left|\iint_{0}^{4\pi }\left[\;{\vec{R}}_{i}\left(\theta ,\varphi \right)\times {\vec{R}}_{j}\left(\theta ,\varphi \right)\right]\;d\Omega \right|}^{2}}{\iint_{0}^{4\pi }{\left|{\vec{R}}_{i}\;\left(\theta ,\varphi \right)\right|}^{2}\;d\Omega \;\iint_{0}^{4\pi }{\left|{\vec{R}}_{j}\left(\theta ,\varphi \right)\right|}^{2}\;d\Omega }$$
where *S_ii_* and *S_ij_* represent the reflection coefficient and transmission coefficients, respectively. ${\vec{R}}_{i}\left(\theta ,\varphi \right)$ and ${\vec{R}}_{j}\left(\theta ,\varphi \right)$ denotes the 3D radiation patterns for the *i*th and *j*th antennas excitation, respectively. Ω implies the solid angle. Figure 13a illustrates the numerical ECC curves for the proposed MIMO antennas. The ECC value is obviously less than 0.045 (radiation patterns)/0.008 (S-parameters), which is substantially lower than the acceptable value (0.5). Thus, the lowered ECC values indicate that the developed MIMO antenna offers an outstanding diversity pattern [56].

### 4.5. Diversity Gain (DG)

DG is another MIMO antenna diversity performance parameter, which presents the antenna diversity effects on the radiated power. The developed MIMO antenna’s diversity gain can be derived from the expression (11), as stated in [55].
(11)$$\mathrm{DG}=10\sqrt{1-{\left|ECC\right|}^{2}}$$

Figure 13b demonstrates the DG plot for the proposed MM-based MIMO antenna. The proposed antenna diversity gain value is higher than 9.96 dB within the 5G NR (n77/n78/n79) spectrum.

### 4.6. Channel Capacity Loss (CCL)

The MIMO antenna system improves the wireless channel capacity, and the CCL indicates the channel capacity loss owing to the correlation effect between the wireless links. Thus, the CCL of the MIMO antenna should be less than the value of 0.4 bps/Hz [57]. The CCL is determined using the following Equations (12) and (13), as stated in [10,57].
(12)$$CLL=-log\;2\;det\;\left(\psi^{r}\right)$$
where $\psi^{r}$ represents the correlation matrix that is found by
(13)$$\psi^{r}=\left[\begin{array}{c} \rho_{11}\;\;\rho_{12}\;\;\rho_{13}\;\;\rho_{14}\\ \rho_{21}\;\;\rho_{22}\;\;\rho_{23}\;\;\rho_{24}\\ \rho_{31}\;\;\rho_{32}\;\;\rho_{33}\;\;\rho_{34}\\ \rho_{41}\;\;\rho_{42}\;\;\rho_{43}\;\;\rho_{44} \end{array}\right]$$
where, $\rho_{ii}=1-\left|\sum_{n=1}^{4}S_{in}^{*}S_{in}\right|$ and $\rho_{ij}=-\left|\sum_{n=1}^{4}S_{in}^{*}S_{nj}\right|$, for i,j=1, 2, 3 or 4.Figure 13c depicts the CCL of the developed 4-port MIMO system with an MM reflector, showing that the CCL value is lower compared to the acceptable value of 0.4 bps/Hz within the working band. Therefore, the designed MIMO system demonstrated an excellent throughput.

### 4.7. Multiplexing Efficiency (ME)

The Multiplexing Efficiency (ME) is another MIMO performance metric that can be derived from Equation (14) [10].
(14)$$ME=\sqrt{\left(1-{\left|\rho \right|}^{2}\right)\eta_{1}\eta_{2}}$$
where ρ denotes the complex ECC and $\eta_{1},\eta_{2}$ denotes the antenna 1 and antenna 2 efficiencies, respectively. Figure 13d shows the ME between the MIMO antennas. It is noticeable that the ME of the designed MM-based MIMO system varies between −1.4 dB and −2.9 dB within the working frequency.

## 5. Performance Comparison with Relevant Works

A comparison of the proposed MM-based antenna system concerning the related state-of-the-art MIMO antenna is tabulated in Table 1. As can be seen, the proposed wideband MM-based MIMO antenna outperforms the recently reported antennas in terms of a high gain and wide operating range, with compact dimensions for 5G NR n77/n78/n79 band applications. Furthermore, the excellent MIMO characteristics of the proposed technique are achieved with the improved isolation between the MIMO components. In [58], the researchers devised a MM-based wideband MIMO antenna for the sub-6 GHz NR spectrum; however, owing to their limited MIMO components, a lower gain with only 0.8 dBi gain enhancement, and lower MIMO characteristics than the proposed antenna, it is not well suited for highspeed advanced 5G applications. Similarly, the developed 4-port MIMO system in [59] covers the 5G NR n78 spectrum; however, they offer a narrow operating frequency band with a low gain and low MIMO performance, which limits its applicability in 5G wireless communications. Although the designed MIMO antenna presented in [12] offers high gain compared to the proposed MIMO antenna, the size of the MIMO antennas is very large, with a much higher edge-to-edge gap (58 mm) and narrow operating range (only n78 band). In addition, the antenna gain is enhanced slightly, by 0.6 dBi, after using the MS technique. On the other hand, the miniature-size MIMO antenna is designed in [59] and [60] with a very limited operating range, low gain, and low MIMO diversity performance. Furthermore, the gap between the antenna and MM in [60] is higher and incompatible with the 5G frequency band. The excellent gain improvement is achieved in our developed MIMO antenna using a wideband MM reflector with a more compact structure than the one reported in [13]. In [13], the authors conceived a MS reflector-based two-port MIMO antenna with a 6 dBi gain improvement, but this antenna offers a narrow bandwidth and a huge dimension, with only two MIMO components. Additionally, the MS reflector is placed far away from the antenna, lacks MIMO performance analysis, and is incompatible with 5G bands, all of which limit its viability for usage in 5G MIMO communications. Thus, the designed MM-based 4-elements MIMO system could be the leading candidate for n77/n78/n79 band applications.

## 6. Conclusions

This paper presents a compact wideband MM-based four-port MIMO antenna with a high gain and wide operating bandwidth for 5G NR sub-6 GHz bands. Four identical fractal slot microstrip antennas are positioned orthogonally in the developed MIMO system, with a miniature size of 0.65λmin × 0.65λmin × 0.02λmin. The designed MIMO antenna operates in the spectrum of 3.25 GHz to 5.6 GHz, covering the 5G NR n77/n78/n79 bands, which is verified both experimentally and computationally. The developed wideband MM reflector is introduced in the MIMO system to enhance the gain and isolation between two close antenna radiators. The proposed antenna reaches a maximum gain of 7.1 dBi, and an isolation of 35 dB, with a 2.8 dBi and 3 dB improvement, respectively, proving the contribution of the MS in the developed MIMO system. The designed MIMO antenna is fabricated and tested, yielding a high peak gain of 7.75 dBi. The measurements confirm the simulated findings with close similarity. Moreover, the MIMO’s diversity characteristics, such as the ECC, DG, ME, and CCL, are evaluated and found to be of outstanding value from the expected limit for 5G MIMO applications. In addition, the designed MIMO system achieved excellent total efficiency (up to 74%) and unidirectional beam patterns by shifting the main beam direction to the top of the radiators, about 60° and 6° in the x-z and y-z planes, respectively. Thus, the proposed antenna shows a promising choice for MIMO applications in the 5G NR n77/n78/n79 spectrums.

## Figures and Tables

**Figure 1 materials-16-01751-f001:**
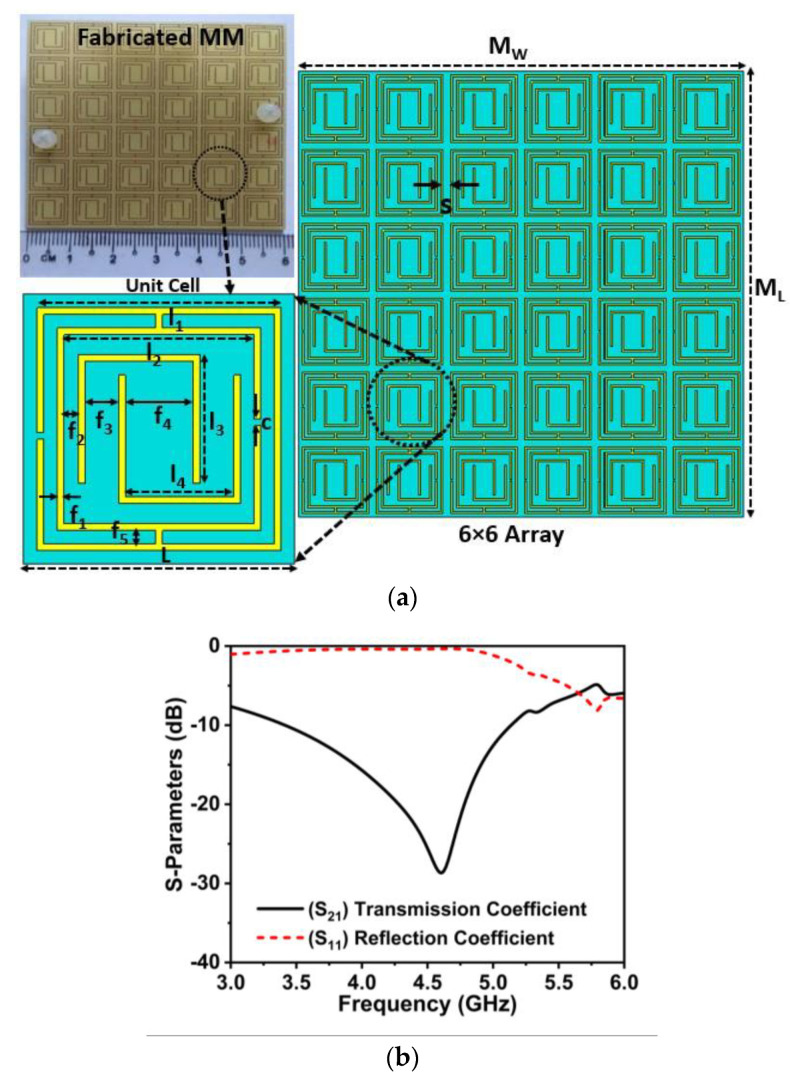
(**a**) MM design details with its fabricated prototype (*L =* 10, *I*_1_ = 9, *I*_2_ = 7.5, *I*_3_
*=* 4.75, *I*_4_
*=* 4.5, *f*_1_
*=* 0.25, *f*_2_
*=* 0.5, *f*_3_
*=* 1.25, *f*_4 *=*_ 2, *f*_5 *=*_ 0.5, *s* = 1.0 and *c* = 0.25, unit: mm) and (**b**) transmission and reflection coefficients curve.

**Figure 2 materials-16-01751-f002:**
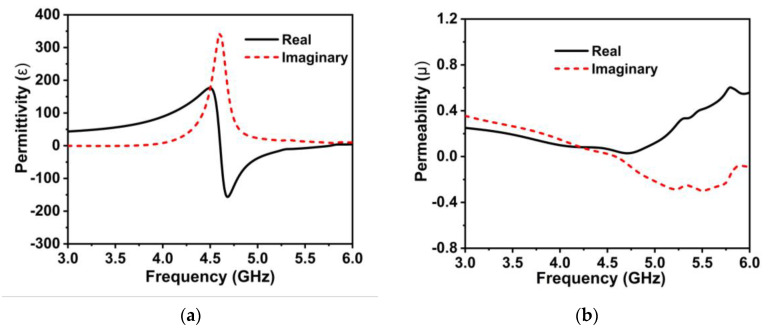
Effective extracted parameters (**a**) permittivity plot, (**b**) permeability plot, and (**c**) refractive index plot.

**Figure 3 materials-16-01751-f003:**
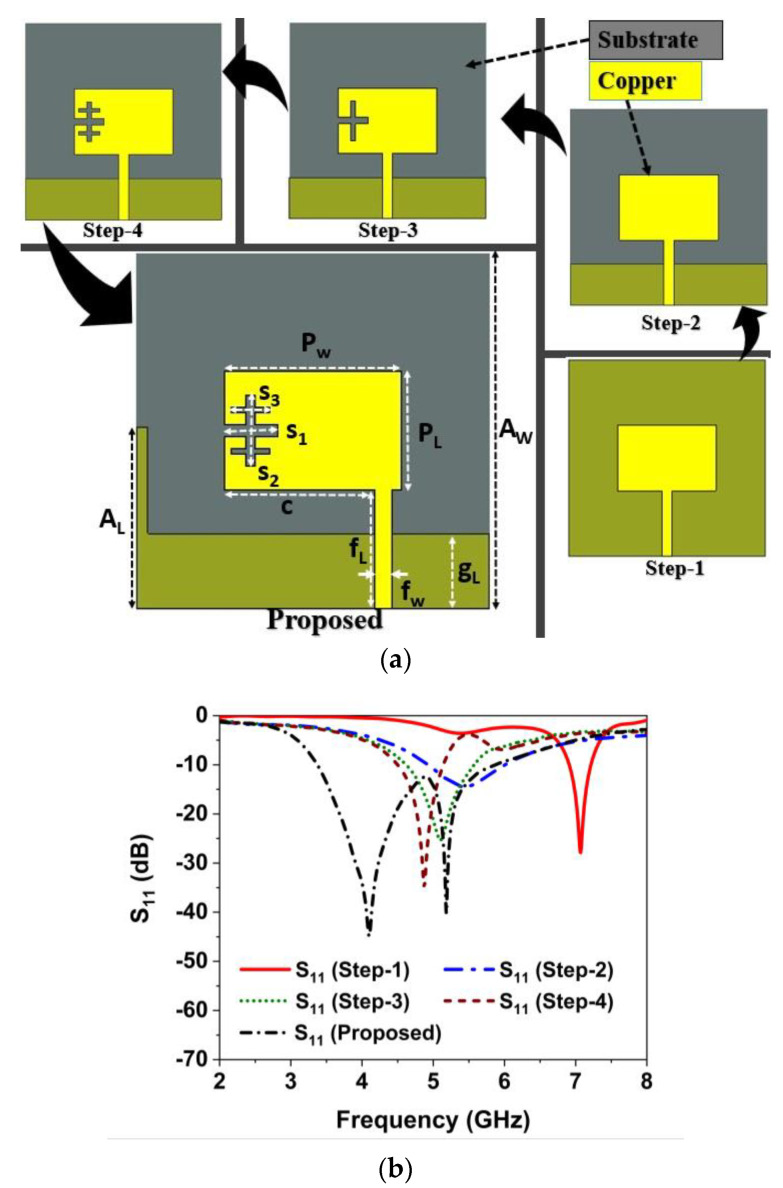
(**a**) Initial antenna design steps (**b**) reflection coefficient (S_11_) results.

**Figure 4 materials-16-01751-f004:**
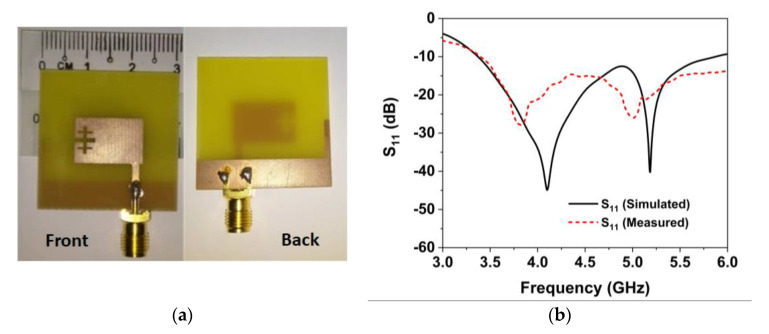
(**a**) Photos of the fabricated single antenna (**b**) Simulated and measured single antenna $\left|S_{11}\right|$ coefficients.

**Figure 5 materials-16-01751-f005:**
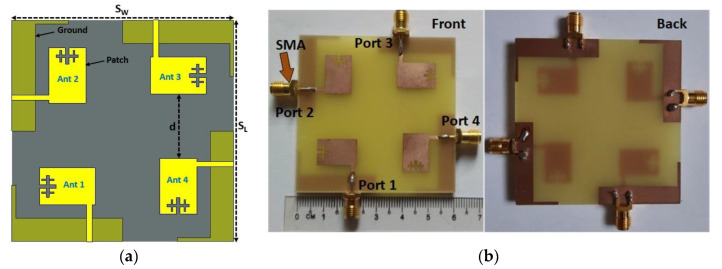
(**a**) Design architecture of the Wideband MIMO antenna (*S_L_*, *S_W_* = 60 mm and *d* = 17.5 mm) and (**b**) fabricated 4-port MIMO antenna prototype.

**Figure 6 materials-16-01751-f006:**
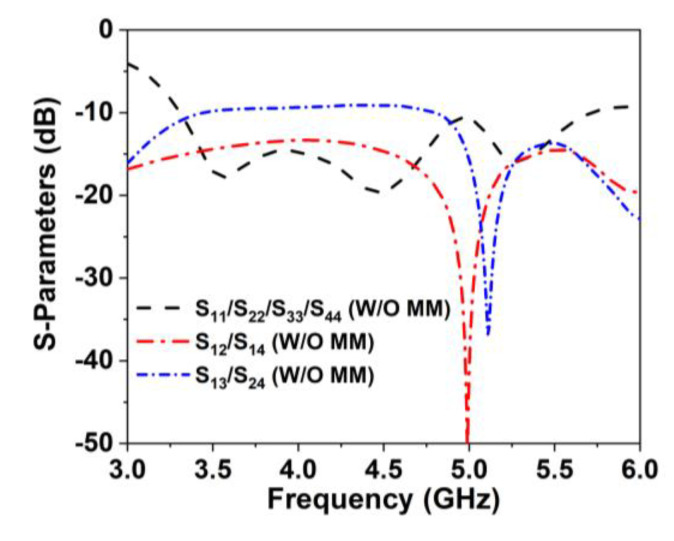
Simulated performance of reflection coefficient (S_11_, S_22_, S_33_, and S_44_) and isolation between MIMO adjacent and diagonal components (W/O: without).

**Figure 7 materials-16-01751-f007:**
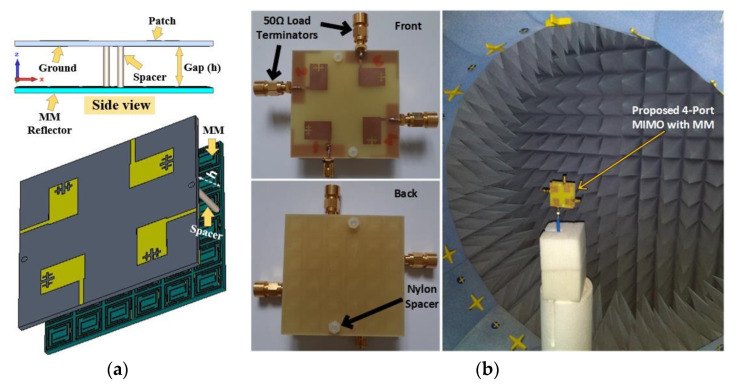
(**a**) Simulation arrangement and side view of the designed MIMO system with MM reflector and (**b**) snapshots of the developed MIMO prototype with MM (front and back) and SATIMO measurement system.

**Figure 8 materials-16-01751-f008:**
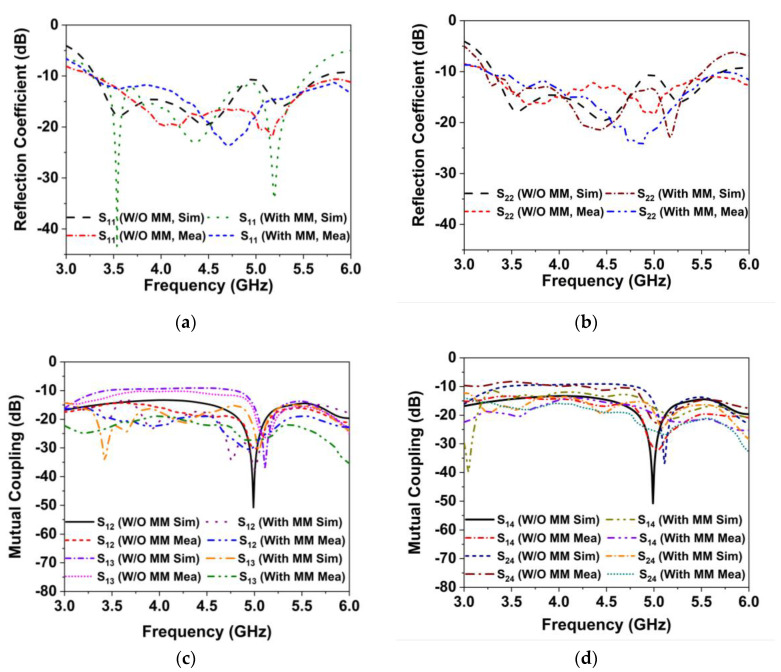
Simulated and tested scattering matrix characteristics of the developed MIMO system without and with MM (**a**) $\left|S_{11}\right|$ (**b**) $\left|S_{22}\right|$ (**c**) $\left|S_{12},\;S_{13}\right|$ and (**b**) $\left|S_{14},\;S_{24}\right|$.

**Figure 9 materials-16-01751-f009:**
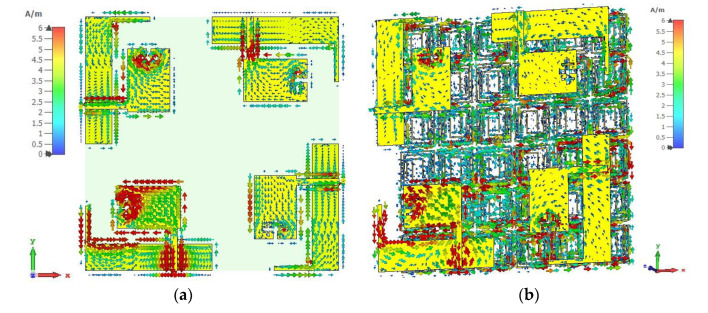
Current distributions at 4.5 GHz for proposed 4-port MIMO antenna (**a**) MIMO without MM and (**b**) MM-based MIMO.

**Figure 10 materials-16-01751-f010:**
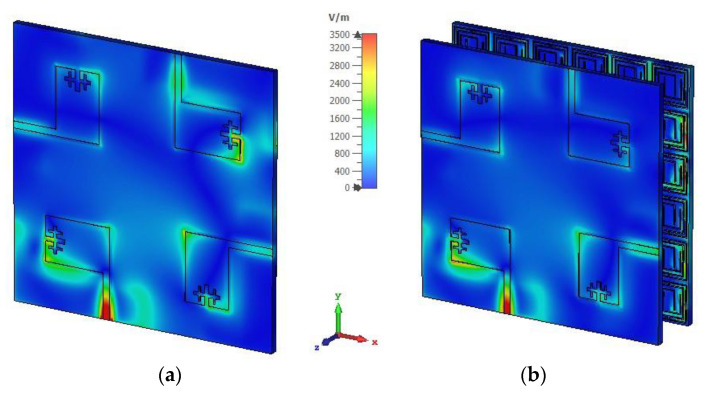
Electric field distributions at 4.5 GHz for proposed 4-port MIMO antenna (**a**) MIMO without MM and (**b**) MM-based MIMO.

**Figure 11 materials-16-01751-f011:**
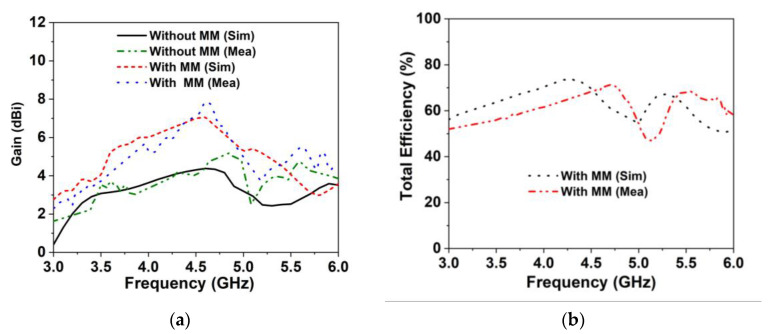
Measured and simulated (**a**) realized gain and (**b**) total efficiency for the developed wideband MIMO antenna.

**Figure 12 materials-16-01751-f012:**
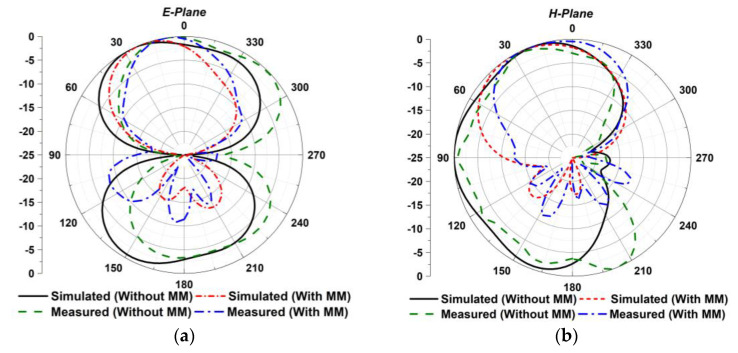
Simulated and measured radiation patterns of the developed MIMO antenna at 4.5 GHz (**a**) E-plane and (**b**) H-plane (without and with MM).

**Figure 13 materials-16-01751-f013:**
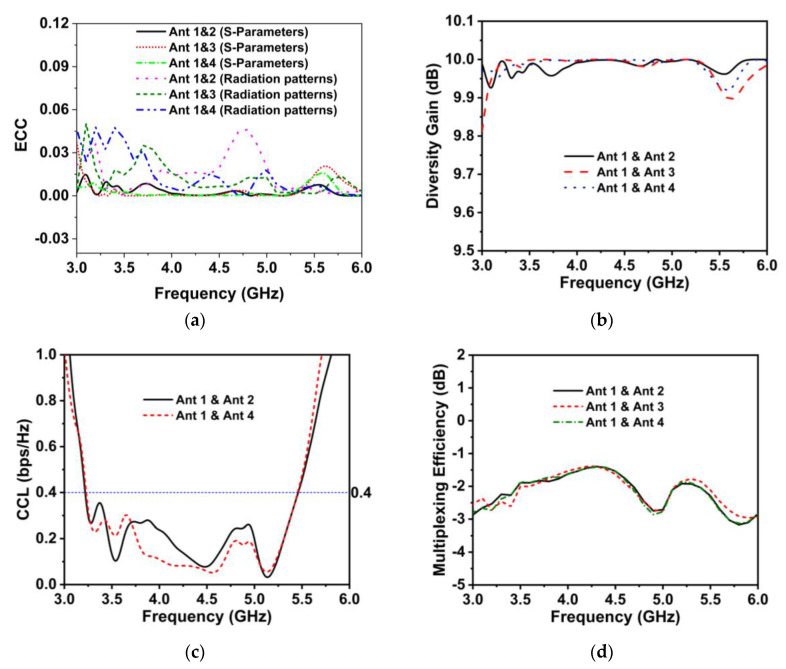
MM-based MIMO diversity characteristics (**a**) ECC, (**b**) DG, (**c**) CCL, and (**d**) ME.

**Table 1 materials-16-01751-t001:** Performance comparison with different relevant antennas.

Ref.	Dimension (λ_min_)	Elements	Antenna Structure	Bandwidth (GHz) (%)	Gain Improvement (dB)	Max. Gain (dBi)	ECC (dB)	DG (dB)	CCL (bps/Hz)
[13]	0.62 × 0.62 × 0.11	2	MIMO + MS	(2.23–2.91) 26.45	6	7.02	Not Given	Not Given	Not Given
[59]	0.38 × 0.98 × 0.008	4	MIMO	(3.26–3.88) 17.42	-	4	<0.10	Not Given	Not Given
[60]	0.46 × 0.46 × 0.19	4	MIMO + MM Reflector	(2.28–2.52) 9.58	-	6	<0.19	Not Given	Not Given
[12]	1.6 × 1.6 × 0.04	4	MIMO + MS	(3.3–3.87) 15.9	0.6	8.72	<0.001	>9.98	Not Given
[58]	0.36 × 0.66 × 0.02	2	MIMO + MM	(3.0–6.0) 70.7	0.8	3.28	<0.02	>9.95	Not Given
This Paper	0.65 × 0.65 × 0.14	4	MIMO+MM	(3.25–5.6) 55.2	2.8	7.10	<0.008 (SP)<0.045 (RP)	>9.96	<0.35

Note: MM: Metamaterial; λ_min_: Wavelength at minimum operating frequency; SP: S-Parameters; RP: Radiation Patterns; Max.: Maximum; Min.: Minimum.

## Data Availability

Not applicable.

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
