# Peer review of "A Compact Mu-Near-Zero Metamaterial Integrated Wideband High-Gain MIMO Antenna for 5G New Radio Applications"

_materials, 2023, doi:10.3390/ma16041751_

Round 1
Reviewer 1 Report
See the following suggestions and also the pdf file
1) Lines 52, 59, 140: I would suggest to specify the used acronym even if detailed in the Abstract
2) Line 77: bandwidth (BW)
3) Line 139: you can add that these investigations are necessary to perform a diversity analysis and say something on diversity. Not all the researchers make the same kind of performance measurements. This can add value to the work
4) Line 140: all the acronyms should be specified- I underlined in yellow
5) Line 142: (EM) the acronym has been specified before. here you can use the entire word
6) Lines 158, 160: use the acronym or the entire word. It is not necessary to repeat both
7) Line 161and line 165: use “shows” instead of “demonstrates”. It is not a demonstration
8) Lines 165 and 167: parameters instead of characteristics
9) Line 172: enlarge symbols in fig. 1a
10) Line 173: reflection
11) Line 177: use "scattering matrix" or "S-parameters", not both
12) Line 179: impedance, z, and refractive index, n,
13) Line 181: specify ko and d
14) Line 185: I suggest the following phrase “From equations (1) and (2), it can be obtained:” in order to not use a sentence very similar to 179line
15) Line 189: explain the significance of (') and ("). probably you can use Imag and Real, instead. Here the real and imaginary part of the operator log is used to extract n
16) Line 190: the sentence in yellow is not clear. I suggest to change as in the following or similarly "where m is the integer denoting the branch index for of the real part of n."
17) Line 196: the same things have been said before. it is better to remove the sentence
18) Line 233: resonates
19) Line 244: remove “which is also shifted to the left”
20) Line 255: the symbols in fig 3a “proposed” are not clear. they need to be enlarged
21) Line 270: coefficients
22) Line 319: the near field (SATIMO)
23) Line 345: these coefficients (S13/S24) are not in the figure 8, or elsewhere
24) Line 355: the lines of the coefficients measured and simulated are not clearly visible. It could be better to divide fig8a and fig 8b each in two parts
25) Line 362-364: from fig. 9b it is not clearly visible how the currents are flowing
26) Line 370-371: the difference in colours, which means the difference in E-field strength, is not clearly visible between fig. 10a and 10b. could you use another CST field graphical representation or change the scales?
27) Line 383: with SATIMO near-field laboratory set-up
28) Lines 387-389: I suggest to shorten the sentence. It is repetitive in the contest
29) Lines 406-408 and 428-430 : the text in yellow (lines 406-408 and 428-430) can be summarized at the end of 4.3 paragraph. For each measurement, it is important to underline the agreement between simulations and measurements but not that your device is suitable for 5G application. something that you say in any case in paragragh 5.
30) Line 474: reflector, showing that the CCL....
31) Line 493: comparison
32) Line 513: a more compact structure than the one reported in [13]

Author Response
Response to Reviewers
Dear Editor,
We appreciate you and the reviewers for your precious time in reviewing our manuscript and providing valuable comments. It was your valuable and insightful comments that led to possible improvements in the current version. The authors have carefully considered the comments and tried their best to address every one of them.
We are uploading (a) our point-by-point response to the comments (below) (response to reviewers), and (b) an updated manuscript with red highlighting indicating changes. Also, the English language has been revised.
Best regards,
Md. Mhedi Hasan et al.

Reviewer 2 Report
Authors have proposed A Compact Mu-near-zero Metamaterial Integrated Wideband High-Gain MIMO Antenna for 5G New Radio Applications. The following are observations / suggestions.
· The abstract mentions the bandwidth, however, it is expected that fractional bandwidth should be specified as wide bandwidth has been claimed in the manuscript. Avoid mentioning “smallest” in the size, you may claim compactness compared to the literature.
· Introduction contains adequate literature. More number of recent publications in the literature may be added.
· The simulator used for the numerical computation may not be necessary kept in quotes.
· The figure 1 needs significant improvement. It is tough to read the contents of both (a) and (b) sub-figures. Also, figure 2 & 3 have similar issues.
· A note should be kept on the dimensions of the metamaterial unit cell. Why this particular dimensions are opted for?
· It is getting difficult to understand the layer-wise bifurcation of the antenna. Could you please improve it for better understanding of end-user?
· The isolation with MM in some cases is poor (Line#289), can you please elaborate the reasons for it?
· It is extremely difficult to assess the current distributions in Figure 9. Please improve the image.
· Please mention steps for further possible improvement in efficiency of the antenna? Can you include a note why efficiency is sub-70%?
· The comparison should have all the dimensions instead of only planar dimensions, preferably expressed in wavelength and bandwidth needs to be in fractional values.
Author Response

(The authors gave the same response as above.)

Round 2
Reviewer 2 Report
Majority of the corrections have been carried out however, the text in majority of the figures are not adequately visible. Please improve it.